# Autoantibody Profiling in Ulcerative Colitis: Identification of Early Immune Signatures and Disease-Associated Antigens for Improved Diagnosis and Monitoring

**DOI:** 10.3390/ijms26094086

**Published:** 2025-04-25

**Authors:** Andreas Weinhaeusel, Jasmin Huber, Silvia Schoenthaler, Florian Beigel, Christa Noehammer, Klemens Vierlinger, Matthias Siebeck, Roswitha Gropp

**Affiliations:** 1Austrian Institute of Technology GmbH (AIT), Giefinggasse, 1210 Vienna, Austria; jasmin.huber@ait.ac.at (J.H.); silvia.schoenthaler@ait.ac.at (S.S.); christa.noehammer@ait.ac.at (C.N.); klemens.vierlinger@ait.ac.at (K.V.); 2Department of Medicine II, Hospital of the Ludwig-Maximilian University Munich, 81377 Munich, Germany; f.beigel@burlefinger-beigel.de; 3Department of General, Visceral und Transplantation Surgery, Hospital of the Ludwig-Maximilian University Munich, 80336 Munich, Germany; matthias.siebeck@med.uni-muenchen.de (M.S.); r.gropp@technologyconsulting.de (R.G.)

**Keywords:** ulcerative colitis, inflammatory bowel disease, autoantibodies, protein microarray, pathways, biomarkers, immune profiling, serotesting, molecular pathology, omics-analysis

## Abstract

Ulcerative colitis (UC) is a major form of inflammatory bowel disease (IBD) characterised by chronic immune-mediated inflammation. While serological biomarkers for IBD diagnosis and differentiation have been explored, autoantibody-based profiling remains underdeveloped. This study aimed to elucidate antibody signatures in manifested and pre-diagnostic UC patients compared to controls using a high-content protein microarray. Serum and plasma samples from manifested and pre-diagnostic UC cohorts were analysed using AIT’s 16k protein microarray, presenting 6369 human proteins. The pre-diagnostic cohort, consisting of 33 UC cases and 33 controls, included longitudinal samples collected before diagnosis, while the severe UC cohort, comprising 49 severe UC patients and 23 controls, included individuals undergoing treatment. Immunoglobulin G (IgG) autoantibody reactivity was assessed to identify differentially reactive antigens (DIRAGs) linked to UC onset, disease progression, and activity. In manifested UC, 691 DIRAGs showed higher reactivity in cases. In the pre-diagnostic cohort, 966 DIRAGs were identified, with 803 antigens exhibiting increased reactivity in cases. Longitudinal analysis revealed 1371 DIRAGs, with 1185 showing increased reactivity closer to diagnosis when comparing samples collected 4–11 months before UC diagnosis to earlier time points 9–24 months prior, highlighting potential early biomarkers. A significant overlap of 286 antigens, corresponding to 41 percent of identified DIRAGs, was observed between severe and pre-diagnostic UC datasets, with an odds ratio of 3.8 and a *p*-value below 2.2 × 10^−16^, confirming reliability and biological relevance. Additionally, 21 antigens correlated with simple clinical colitis activity index (SCCAI) scores. Reactome pathway analysis identified 49 pathways associated with DIRAGs in pre-diagnostic UC, distinct from 24 pathways in manifested UC, with an overlap of five key pathways related to protein folding, immune regulation, and viral infection, reflecting differences in disease onset and manifestation. Autoantibody profiling reveals early immune signatures in UC, offering novel biomarkers for preclinical diagnosis and disease monitoring. The overlap between pre-diagnostic and manifested UC antigenic profiles reinforces their biological relevance, linking them to molecular pathology. These findings highlight antibody profiling as an additional omics layer, paving the way for new diagnostic and therapeutic strategies in UC management.

## 1. Introduction

Ulcerative colitis (UC), a leading form of inflammatory bowel disease (IBD), imposes a substantial burden on individuals and healthcare systems globally. The disease is characterised by chronic mucosal inflammation confined to the colon and rectum. Although the precise aetiology remains elusive, UC is thought to result from a complex interplay of genetic predispositions, environmental triggers, and immune dysregulation targeting the gut microbiota [1,2]. Notably, UC disproportionately affects individuals in developed countries, with most cases diagnosed between the ages of 30 and 40 [3].

Diagnosis of UC is often delayed, relying on invasive endoscopic examinations and histopathological analysis of biopsies. These methods, while effective, are burdensome for patients and contribute to delayed therapeutic interventions. Serological biomarkers offer a promising alternative by enabling earlier detection through less invasive means. Among these, autoantibody profiling holds potential not only for diagnostic purposes but also for elucidating the pathophysiological underpinnings of UC [4,5].

Recent studies have highlighted the role of autoantibodies in the pathogenesis and progression of UC, suggesting that these immune markers may serve as both diagnostic tools and potential therapeutic targets. Autoantibodies, aberrant immune proteins directed against self-antigens, are reflective of immune dysregulation. Previous studies have identified UC-associated autoantibodies, such as those targeting perinuclear antigens (pANCA) and epithelial proteins [6,7]. However, a comprehensive profiling of autoantibody responses to elucidate their diagnostic and prognostic relevance has remained largely unexplored. Early detection of UC-associated autoantibodies could serve as a critical adjunct to current diagnostic paradigms, aiding in disease stratification and therapeutic monitoring [8].

In our previous paper [9], we showed the relevance of autoantibody detection using a high-density protein microarray. We systematically profiled Immunoglobulin G (IgG) autoantibody responses to identify robust biomarkers, such as the transmembrane protein cluster of differentiation 99 (CD99). Focusing on CD99 as an antigenic biomarker, we have found in vivo that CD99 exposure in NOD/ScidIL2Rγnull (NSG-UC) mice reconstituted with UC donor peripheral blood mononuclear cells (PBMCs) exacerbated inflammation and disease severity. Here, we focus on the protein-array data analysis and go into these details.

The overarching goals of this research are twofold: (1) to evaluate the utility of autoantibody profiling in detecting UC at preclinical stages, and (2) to provide mechanistic insights into the immune dysregulation underlying UC. By integrating data from severe and pre-diagnostic cohorts, we aim to establish a comprehensive framework for using autoantibody profiles as a diagnostic tool and to show that these could provide novel therapeutic options. The current study supports emerging evidence that autoantibodies may precede the clinical onset of UC by several years [10], and aims to map immune signatures correlating with disease progression and activity. In doing so, this study not only advances our understanding of UC pathogenesis but also paves the way for personalised patient care.

## 2. Results

In our study, we analyse antibody profiles using protein microarrays in two distinct cohorts: severe UC patients vs. controls and pre-diagnostic samples collected prior to UC diagnosis, along with matched controls. This approach enables us to identify differentially reactive antigens (DIRAGs) associated with severe UC and define those linked to disease activity. Additionally, we determine DIRAGs in pre-diagnostic UC by comparing these samples against controls and conducting paired analyses of pre-diagnostic samples collected closest to diagnosis (6 months before) vs. samples obtained approximately 21 months prior to diagnosis. Furthermore, we investigate the overlap of DIRAGs between severe UC and pre-diagnostic UC, identifying potential biomarkers for early diagnosis. Finally, we explore the biological relevance of these antibody profiles through Reactome pathway analysis, providing insights into their potential role in UC pathogenesis.

### 2.1. Antibody Profiles in Severe UC vs. Controls

To pursue a comprehensive analysis of autoantibodies in UC vs. controls, blood samples were collected at LMU from 49 patients already classified as severe UC cases and 23 non-affected individuals, as shown in Table 1. This set enables the analysis of antibody profiles in UC vs. controls, as well as to find associations by correlation analysis with the clinical disease activity index (SCCAI).

For conducting protein array analysis, we purify IgG from serum, perform a protein concentration measurement, and load standardised amounts of IgG onto protein arrays. This IgG purification step enables us to analyse the IgG protein concentrations in samples. For the IgG concentration measured, we have found a statistically significant difference (*p* = 0.0027) of IgG amounts/concentrations in serum from UC cases (median = 10.64 mg/mL serum) vs. controls (8.86 mg/mL). To test whether disease activity affects IgG concentrations, we performed a correlation analysis. The Pearson correlation coefficient between IgG levels (mg/mL) and the SCCAI Score was −0.162, with a *p*-value of 0.266. This indicates a weak negative correlation, which is not statistically significant. Therefore, no strong evidence was found to suggest that disease activity has a direct impact on IgG concentrations. This is further supported by the fact that no significant differences in IgG concentrations were observed when comparing UC cases with mild symptoms, those in remission, or when comparing patients with colectomy vs. those without. These findings suggest that IgG concentrations are not significantly influenced by disease activity or surgical intervention in UC.

### 2.2. Protein Array Class Comparison Results of Manifested UC vs. Controls

When loading a standardised IgG amount of 67.5 µg onto microarrays, class comparison analysis between cases and controls (49 UC patients and 23 non-affected individuals, see Table 1) elucidated 691 significant DIRAGs at the nominal *p* < 0.01 level (Appendix A). In our previous paper [9], we showed the relevance of CD99 as a biomarker and found additional in vivo evidence that CD99 exposure in NSG-UC mice reconstituted with UC donor PBMCs exacerbated inflammation and disease severity. Here, we focus on the protein array data analysis and go into these details.

The top list of 20 DIRAGs sorted by fold change (FC) between classes, elucidates proteins like DMPK, GGA, HES, RPS29, and CD99 with higher antigenic reactivities in cases (Table 2). Of the 691 significant antigens between cases and controls, 63 antigenic proteins were listed ≥2 times and thus expressed from different clones presented on arrays.

### 2.3. Antibody Reactivities Correlated with Disease Activity

To investigate antibody reactivities against antigenic proteins associated with disease activity in 43 UC patients, we conducted a correlation analysis. Using median-normalised data, we identified 26 proteins significantly correlated with disease activity scores (*p* < 0.01). The Pearson correlation coefficients ranged from −0.444 to −0.365. Given the potential influence of median normalisation on skewing the original data, we repeated the correlation analysis using unnormalised protein array data. This analysis revealed 21 antigens significantly correlated with the SCCAI (*p* < 0.01), with Pearson correlation coefficients ranging from −0.57 to −0.456. These results are summarised in Table 3. When comparing the intersections of the results from both normalised and unnormalised datasets, we identified three proteins (CLTA, ZWINT, and TUFM) that were consistently present in both analyses.

#### Antibody Reactivities Correlated with Treatments

To explore treatment-related antigenic reactivities, we performed ANOVA models incorporating main effects and interaction terms for Mesalazine, Cortisone, and TNFα-blockers; treatment of patients is summarised in a nested table in Appendix A. Notably, several interaction terms were associated with differential antigen expression at a nominal significance threshold (*p* < 0.01). The three-way interaction (Mesalazine: Cortisone: TNFα-blocker) yielded the highest number of nominally significant hits (*n* = 22), followed by the interactions Mesalazine: Cortisone (*n* = 18), Mesalazine: TNFα-blocker (*n* = 17), and Cortisone: TNFα-blocker (*n* = 10). Although none of these findings passed FDR correction, they suggest potential combinatorial treatment effects on immune-related antigen expression, warranting further investigation in larger cohorts. We have summarised different treatment regimens, as well as the ANOVA results and corresponding lists of antigens in Appendix A, respectively.

Combining all antigens identified as significant across the ANOVA models yielded a total of 41 unique antigens. When these were subjected to pathway enrichment analysis using WebGestalt, seven pathways were found to be significantly enriched (*p* < 0.01; see Appendix A). These include the immune response-regulating signalling pathway (GO:0002764), non-canonical NF-kappaB signal transduction (GO:0038061), positive regulation of defence response (GO:0031349), translational initiation (GO:0006413), protein kinase A signalling (GO:0010737), positive regulation of response to biotic stimulus (GO:0002833), and positive regulation of cytokine production (GO:0001819). While diverse in nomenclature and specific molecular functions, these pathways converge on several key processes related to immune system activation, intracellular signalling, and cellular responses to external stimuli. This suggests that the antigenic profile captured in our study reflects a coordinated immune response potentially shaped by therapeutic interventions.

### 2.4. Antibody Profiles in Pre-Diagnostic UC

Pre-diagnostic UC samples and matched controls were analysed to identify antibody reactivity changes before clinical manifestation and compare DIRAGs/antibody profiles in early and manifested UC.

The pre-diagnostic study on UC utilised plasma samples sourced from the Biobank of the Bavarian Red Cross (BRK), which hosts a repository of 400,000 active blood donors. Looking up the biobank revealed that among the entire collection, 11 pre-diagnostic cases were available and of these 9 individuals’ samples could be found for which a sample was available closest to clinical UC diagnosis (T3: median 6 [range 1–11] months before) and at T-early (depending on the available time points T1 or T2 of samples from the individuals) at a median 21 months before UC diagnosis. Thus, the median timespan between T3 and T-early is 13 (9–24) months. These nine individuals were thus available for paired testing. For all these single samples, matched healthy controls have also been provided by the BRK biobank and were analysed together with the pre-diagnostic UC samples using the antibody profiling approach on the AIT-16k protein arrays.

This structured approach allowed the evaluation of autoantibody profiles over time, providing insight into immune changes preceding the onset of UC. Samples that met the criteria formed the foundation for investigating biomarkers associated with early immune dysregulation.

#### 2.4.1. Plasma IgG Concentrations in Pre-Diagnostic UC Samples Do Not Increase Until Diagnosis

In this setting, we analysed 66 samples comprising 33 samples from 11 UC patients derived from three different time points prior to diagnosis and 33 healthy controls matched for gender, time, and centre of blood collection. Sample characteristics of these pre-diagnostic UC patients are summarised in Table 4.

Plasma IgG concentrations in patients (considering T3 closest to diagnosis; *n* = 11) obtained a median IgG serum concentration of 7.31 mg/mL (SD = 1.29) vs. controls (median = 7.77 mg/mL, SD = 1.44; *n* = 33) were not significantly different. Paired analysis of serum concentrations in UC at different time points was then conducted, considering eight individuals when paired samples were available within a year (12 months) prior to UC diagnosis and comparing the IgG serum concentrations with samples taken 9–24 (median 13 months before T3). In eight patients, comparing IgG plasma concentrations at T3 vs. samples taken 9–24 months (median 13 months) before T3, there is a 6,8% median increase in these eight UC cases. However, the difference in serum IgG between time points is not significant (*p* = 0.938; paired T-Test).

Therefore, we conclude that IgG plasma concentrations in pre-diagnostic UC samples do not increase until diagnosis, but changes are affecting the reactivity profiles, then targeting self-antigens.

Although a direct comparison of IgG concentrations from study groups (severe UC case/control serum; pre-diagnostic BRK case/control plasma) is not advisable due to the systematic difference between serum vs. plasma samples. Based on the data described above, IgG concentrations might increase during the progression into severe UC, as seen when serum IgG concentrations were significantly higher in severe cases vs. controls. Whereas in the pre-diagnostic study cohort, the plasma IgG concentrations do not show significant changes in (1) cases vs. controls as well as (2) in cases at time points “closest to” vs. “>2 years before” UC diagnosis.

#### 2.4.2. Differentially Reactive Antigens in Pre-Diagnostic UC

As indicated above, of the pre-diagnostic UC samples, nine individuals were selected (Table 4) for comparative analysis of a sample closest to clinical diagnosis (<12 months) and a sample derived from an earlier time point (median 13 months before T3 sample collection). When applying class comparison in a paired manner, 1371 features have been identified and show a similar pattern as in manifested UC (see Figure 1A). Most DIRAGs exhibit higher reactivity in UC, as illustrated in the volcano plot (see Figure 1B) and shown in detail in Appendix A.

The FC values indicate higher autoantigenic reactivity in samples closest to clinical UC diagnosis (time point T3, median 6 months before clinical diagnosis, 1167 antigens) vs. samples taken earlier (median T-early 13 months, 204 antigens). In total, 1167 antigens show higher reactivity at T3, while 204 are more reactive at T-early. The 20 antigenic proteins with the highest FC and increased reactivity at T3 are listed in Table 5.

#### 2.4.3. Potential for Early Diagnostics

As shown in the class comparison approach and illustrated by volcano plots, UC development seems to be associated with or even driven by a general and broad autoantigenic reaction. When increased antigenic reactivities are affecting a broad spectrum of different human proteins, the elucidation of a minimal set of candidates for obtaining the best classification success has been conducted using class prediction analysis of nine paired pre-diagnostic samples. Although the sample size is limited, the paired analysis improves statistical power. By applying different feature selection strategies and classification algorithms, we achieved a correct classification rate of 78%. Specifically, seven out of nine paired pre-diagnostic IgG profiles were correctly classified when comparing patient samples taken close to diagnosis (4–11 months prior, median 6 months) with those collected at an earlier time point (median 21 months before diagnosis; 9–24 months prior to the last available plasma sample from the BRK blood bank). These earlier samples were provided when the individuals were considered “healthy” blood donors, despite the presence of autoantibodies. The top features of 11 antigens (DCAF5, PPID, GBAS, LRP5, PKM, WNK2, POGLUT1, AAAS, LMO4, AP2S1, SLC22A17) are listed in Table 6, which have an FC ≥ 1.5 between the two time points studied. Of note, samples of the two patients who failed correct classification were among the oldest and longest stored samples (collected 2003–2004 and stored at the BRK blood bank in Munich). Analysing the remaining seven pairs gave rise to more convincing results by defining 56 antigens which enable 100% correct classification of the samples closest to clinical diagnosis vs. those collected about 2 years before. The corresponding results are listed in Table 7 (sorted by FC; FC ≥ 1.6; *p* < 5 × 10^−4^, univariate misclassification rate FDR < 1 × 10^−6^). Although these findings are statistically underpowered, the candidates defined here would be of high interest for further validation on independent serum or plasma samples if available.

In respect to “blood donations and transfusion medicine”, one could speculate that blood derivatives might be causative for unwanted side reactions in recipients using blood derivatives from such donors. Exclusion of blood donors or exclusion of obtained donations based on autoantibody profiles, however, might not be feasible from an organisational and economic point.

#### 2.4.4. Comparative Analysis of Autoantibodies in Manifested UC (Serum) vs. Pre-Diagnostic UC (Plasma)

To assess the quality and reliability of antibody profiles identified in both (1) the “severe UC case/control” study and (2) the “pre-diagnostic UC samples” study (“T-early vs. T3 at diagnosis”), we analysed antigen data across these datasets. For the severe UC case/control study, we examined a set of 691 antigens and applied this list to the pre-diagnostic UC dataset. A class comparison of “cases at T3” vs. “controls” identified 136 significant antigens (odds ratio: 1.9244; 95% CI: 1.5705–2.3581; chi^2^ *p*-value = 2.203 × 10^−10^; Appendix A). Similarly, when analysing the 691 antigens in a paired class comparison of “T-early vs. T3 at diagnosis” within the pre-diagnostic UC dataset, 210 antigens were found to be significant (odds ratio: 3.4231; 95% CI: 2.8593–4.0980; chi^2^ *p*-value < 2.2 × 10^−16^; Appendix A). To refine the findings, we intersected the 1371 antigens from the pre-diagnostic UC dataset with the 481 significant antigens identified in the severe UC vs. controls dataset (upon applying a more stringent filtering criterion of array data by excluding proteins with a mean intensity value below 500, we identified 481 significant antigens between cases and controls; *p* < 0.01; FC ≥ 1.35). This comparison resulted in 196 common antigens out of 481, a highly significant overlap (odds ratio: 6.4088; 95% CI: 5.3011–7.7479; chi^2^ *p*-value < 2.2 × 10^−16^).

Further, excluding two misclassified patients in a paired analysis nearly doubled the number of significant antigens to 2479 (Appendix A). When intersecting this expanded set with the 691 antigens from the severe UC vs. controls dataset, we identified 286 common antigens, again demonstrating a highly significant overlap (odds ratio: 3.7866; 95% CI: 3.2355–4.4315; chi^2^ *p*-value < 2.2 × 10^−16^).

Overall, 286 antigens were consistently present in both the pre-diagnostic and severe UC study settings, reinforcing the robustness of our findings. Given the highly significant overlap, we conclude that the identified antigens and findings in both comparative case/control studies of severe UC and pre-diagnostic UC are highly reliable. However, while changes in autoantibody profiles during disease progression may contribute to some differences, we must also consider the systematic impact of pre-analytical conditions, particularly the use of serum vs. plasma, on IgG profiles.

#### 2.4.5. Pathway Analysis of Autoantibodies in Manifested UC (Serum) and Pre-Diagnostic UC (Plasma)

Using the lists of significant antigens in our severe UC study and pre-diagnostic settings, we conducted Reactome pathway analysis using the Webgestalt online tool (https://www.webgestalt.org/, accessed on 31 January 2025). Therefore, we defined the entire list of antigenic proteins presented on the AIT 16k protein array as the background list, corresponding to 6125 single antigens.

We have chosen over-representation analysis (ORA) and set an FDR < 0.05 to identify Reactome pathways. Both lists of DIRAGS (using gene symbols) derived from the class comparison analysis were subjected to this Webgestalt analysis and we found 24 GeneSets in (1) the “severe UC case/control setting” (using the list of 691 antigens; pathways shown in Table 8A) and 40 pathways in (2) the “pre-diagnostic UC samples” study (“T-early vs. T3 at diagnosis”; using the list of 1371 antigens; shown in Table 8B). Overlapping pathways found in both sets are shown in bold in Table 8A,B. These comprise the pathways R-HSA-389960, formation of tubulin folding intermediates by CCT/TriC; R-HSA-389958, cooperation of prefoldin and TriC/CCT in actin and tubulin folding; R-HSA-9646399, aggrephagy; R-HSA-5663205, infectious disease; and R-HSA-9824446, viral infection pathways.

Thus, we conclude that the identified antigens and pathways in both comparative case–control studies of severe UC and pre-diagnostic UC reflect the differences between disease onset and manifestation. While changes in autoantibody profiles during disease progression may contribute to these differences, it is important to consider that variations in pre-analytical conditions, particularly the use of serum vs. plasma, may also systematically influence IgG profiles. In Section 3, we further examine these findings in relation to the current literature, providing insights into their molecular pathological interpretation and their relevance to disease biology.

## 3. Discussion

For discussing our results from antibody profiling of UC patient samples on protein arrays, we used the deduced lists of significant DIRAGs and sorted them according to higher reactivity in manifested UC disease (UC vs. controls derived from serum samples) as well as higher antigenic reactivity in plasma samples from patients closest to UC manifestation (6–12 months prior) compared to samples collected more than 12 months before.

We systematically reviewed the literature and searched for potential molecular pathological associations of these top-ranked antigens with UC. Similarly, we analysed the pathways deduced from WebGestalt analysis for both the “manifested UC” and “pre-diagnostic UC” experimental settings.

### 3.1. Discussion of Top Antigens from Class Comparison and Correlation Analysis with Disease Activity

We systematically reviewed the literature to identify potential molecular pathological associations of these top-ranked antigens with UC. Accordingly, we first discuss the top antigens ranked by FC in manifested UC, followed by those correlated with disease activity, and finally, we examine the top antigens identified in pre-diagnostic UC.

#### 3.1.1. Top Autoantigens in Manifested UC

To analyse the top 20 autoantigens sorted by the FC with higher reactivity in UC, which were derived from the class comparison of manifested UC vs. controls (listed in Table 2), we explore their potential roles in inflammation, disease, and specifically UC. As shown below, we have also identified common features or roles among these autoantigens, particularly focusing on those that may contribute to inflammatory processes.

Table 2 includes the autoantigens DMPK, HES1, GGA1, RPS29, CD99, TRIM27, MCM3AP, POGZ, SLC16A8, FLAD1, NBPF9, SCAP, IP6K2, POLR3H, RPS17, UBXN4, YES1, AP2S1, and SEPTIN7P14. CD99, HES1, TRIM27, SLC16A8, MCM3AP, and IP6K2 are found in the literature with inflammation and known disease association: CD99 is a cell adhesion molecule that plays a significant role in leukocyte migration and adhesion, which are critical processes in inflammation. Its involvement in immune responses makes it particularly relevant in inflammatory diseases, including UC [11]. HES1 encodes a transcription factor involved in the Notch signalling pathway, which is crucial for cell differentiation and proliferation. Dysregulation of Notch signalling has been implicated in various inflammatory diseases and cancers, suggesting that HES1 may play a role in the inflammatory processes associated with UC [12]. TRIM27 is part of the tripartite motif (TRIM) family and is involved in various cellular processes, including immune responses and inflammation. TRIM27 has been shown to regulate the NF-κB signalling pathway, which is a key player in inflammation [13]. Its role in modulating immune responses could be significant in the context of UC. SLC16A8 encodes a monocarboxylate transporter that is involved in the transport of lactate and pyruvate across cell membranes. Alterations in metabolic pathways, including those involving SLC16A8, can influence inflammation and metabolic dysregulation, which are relevant in UC [14]. MCM3AP is associated with DNA replication and has been linked to cell cycle regulation. Dysregulation of the cell cycle can lead to increased cell turnover and inflammation, which are characteristic of UC [15]. IP6K2 is involved in inositol phosphate metabolism. IP6K2 has been shown to play a role in cellular signalling and has been implicated in inflammation and immune responses. Its involvement in these processes suggests a potential link to UC [16].

The common features and roles of the autoantigens in Table 2 can be categorised based on their involvement in inflammation and disease processes. Many play key roles in cellular signalling pathways that regulate immune responses, cell adhesion, and metabolic functions. For example: (A) Cell Adhesion and Migration: CD99 and TRIM27 are involved in immune cell interactions and migration, which are critical in the inflammatory response. (B) Cell Cycle Regulation: HES1 and MCM3AP are linked to cell proliferation and differentiation, highlighting the importance of cell cycle regulation in inflammation. (C) Metabolic Regulation: SLC16A8 and IP6K2 are associated with metabolic pathways that can influence inflammatory states, indicating that metabolic dysregulation may contribute to the pathogenesis of UC.

Thus, all in all, many of the autoantigens listed are interconnected through their roles in inflammation and disease processes. Proteins such as CD99, HES1, TRIM27, and SLC16A8 highlight a common theme of involvement in immune regulation and inflammatory responses, which are particularly relevant in the context of UC. Understanding these connections may provide insights into potential therapeutic targets for managing inflammation-related diseases.

#### 3.1.2. Antigens Found Correlated with Clinical Activity Score

To analyse the autoantigens listed in Table 3A (derived from median-normalised data) and Table 3B (derived from unnormalised data, based on the assumption that loading standardised IgG amounts onto arrays is sufficient for standardisation and prevents potential skewing by median normalisation), we will first examine each table individually. Our focus will be on their potential roles in inflammation, disease, and specifically UC. Following this, we will discuss the overlapping genes and proteins—CLTA, ZWINT, and TUFM—found in both tables, highlighting their common features and roles.

Table 3A includes the following autoantigens: PODXL2, FBLL1, PHF19, PRPF3, CLTA, TMEM44, CHAMP1, TTLL7, CLEC11A, PIPSL, ATXN10, KNDC1, ZWINT, TUFM, USP34, SUPV3L1, A2M, PAICS, PDIA4, CCT8, XRCC6, ROBO3, OLA1, BSG, PRMT2, and MEIS3. Many of these have been implicated in inflammatory processes and various diseases. For instance, CLTA (clathrin–AP180 complex) is involved in endocytosis and has been linked to immune responses, suggesting a role in inflammation [11]. ZWINT is associated with the regulation of the cell cycle and has been implicated in cancer progression, which often involves inflammatory pathways [12]. TUFM (Tu translation elongation factor mitochondrial) has been shown to modulate inflammation through its interactions with NLRX1, a key regulator of immune signalling [13]. Moreover, several proteins in Table 3A, such as A2M (alpha-2-macroglobulin), are known for their anti-inflammatory properties and roles in tissue repair [14]. The presence of proteins like XRCC6, which is involved in DNA repair, indicates a potential link to inflammatory responses during cellular stress or damage, common in UC [15].

Table 3B consists of the following autoantigens: DCAF13, ACO2, C17orf70, TPP2, RPL7A, VARS2, RAB5C, DHX30, PTPN6, MINA, CNTD1, MCM6, ZWINT, RSRP1, GOLGA2, CLTA, TUFM, CCDC94, HSF1, CCT2, and FLNA. Similar to Table 3A, several antigens in Table 3B are associated with inflammation and disease. For example, MCM6 is involved in DNA replication and has been linked to cancer and inflammatory responses [16]. CLTA and TUFM, as discussed previously, also play significant roles in immune responses and inflammation [13]. Additionally, PTPN6 (protein tyrosine phosphatase, non-receptor type 6) is known to regulate immune cell signalling and has been implicated in various inflammatory diseases [17].

The overlap of autoantigens between Table 3A and B, particularly CLTA, ZWINT, and TUFM, suggests a shared functional significance in inflammation and disease processes.

CLTA is crucial for endocytosis, a process that is vital for immune cell function and the regulation of inflammation. Its role in the internalisation of receptors can modulate inflammatory signalling pathways, making it a key player in the immune response [11]. In the context of UC, dysregulation of endocytic pathways may contribute to the pathogenesis of the disease.

ZWINT is involved in the regulation of the cell cycle and has been shown to interact with proteins that are critical for mitosis. Its role in cancer progression highlights its potential involvement in inflammatory responses, as cancer and inflammation are closely linked [12]. In UC, the dysregulation of cell cycle genes can lead to increased cell turnover and inflammation.

TUFM is a mitochondrial protein that plays a significant role in protein synthesis and has been implicated in the regulation of autophagy and inflammation. It interacts with NLRX1 to modulate inflammatory responses, suggesting that it may play a protective role against excessive inflammation [13]. In UC, where inflammation is a hallmark, TUFM’s regulatory functions could be crucial for maintaining intestinal homeostasis.

In summary, the lists of antigenic proteins found correlated with disease activity scores reveal that many of the listed autoantigens are interconnected through their roles in inflammation and disease processes. The overlapping autoantigens—CLTA, ZWINT, and TUFM—highlight a common theme of involvement in immune regulation and inflammatory responses, which are particularly relevant in the context of UC. Understanding these connections may provide insights into potential therapeutic targets for managing inflammation-related diseases.

#### 3.1.3. Top Autoantigens in Pre-Diagnostic UC

The list of autoantigens in Table 5 includes the following proteins: DCAF5, SRSF9, LAMC1, SULT1A3, EVL, CKAP5, TINF2, DISP3, NAP1L4, PTPRE, EIF2S3, TIAL1, FAM13A, USP11, FMNL2, HNRNPA2B1, PPID, MAGED2, SUGP2, and FAU. Here, we summarise potential associations with respect to UC of the top three autoantigens and selected others.

DCAF5: This protein is involved in the regulation of protein degradation and has been linked to cellular stress responses. Its role in modulating protein stability may influence inflammatory pathways, although specific references to its involvement in UC are limited.SRSF9: This gene encodes a splicing factor that plays a role in RNA processing. Alterations in splicing can affect the expression of inflammatory mediators, suggesting a potential link to inflammatory diseases, including UC [11].LAMC1: As a component of laminin, LAMC1 is crucial for cell adhesion and maintaining the integrity of the extracellular matrix. Disruption in LAMC1 expression can lead to impaired epithelial barrier function, which is particularly relevant in UC, where barrier integrity is compromised [12].

USP11 is a deubiquitinating enzyme that regulates protein stability and inflammatory signalling pathways. Its involvement in modulating TGF-β signalling suggests a role in inflammation [13]. HNRNPA2B1, a heterogeneous nuclear ribonucleoprotein, is involved in RNA processing and has been implicated in the regulation of inflammatory gene expression, linking it to the pathogenesis of inflammatory diseases [14]. CKAP5 is associated with cytoskeletal organisation and plays a role in immune cell function and migration during inflammation, which may be relevant in UC.

The autoantigens shown in Table 6 (top features sorted by FC derived from paired analysis of nine patient pairs (T3 vs. T-early)) share functional similarities with those in other tables. For instance, CD99 is also involved in cell adhesion and immune responses, paralleling the role of LAMC1 in maintaining epithelial integrity. Similarly, TRIM27, which regulates NF-κB signalling, has a complementary role to USP11 in modulating inflammatory pathways.

The analysis reveals that many of the autoantigens listed are interconnected through their roles in inflammation and disease processes. Antigens such as LAMC1, USP11, and HNRNPA2B1 highlight a common theme of involvement in immune regulation and inflammatory responses, particularly relevant in the context of UC. Gaining insight into these connections could help identify potential therapeutic targets for treating inflammation-related diseases.

### 3.2. Discussion of Selected Pathways

The pathogenesis of UC is a complex interplay of genetic, immunological, and environmental factors. Recent analyses have identified several key Reactome pathways that contribute to the disease’s manifestation and progression. This discussion will focus on the roles of the top three pathways derived from analyses of manifested disease vs. controls, followed by an exploration of pathways overlapping in both manifested and pre-diagnostic UC, and finally, the pathways identified in pre-diagnostic UC prior to clinical manifestation.

#### 3.2.1. Top 3 Pathways in Manifested Ulcerative Colitis

Vesicle-mediated transport (R-HSA-5653656) plays a crucial role in the secretion and recycling of proteins and lipids essential for maintaining the integrity of the intestinal epithelium. Disruptions in vesicular transport can lead to impaired epithelial barrier function, which is a hallmark of UC. Studies indicate that altered vesicle trafficking can exacerbate inflammation by affecting the delivery of immune mediators and contributing to the dysregulation of intestinal homeostasis [18,19]. Furthermore, the involvement of vesicle-mediated transport in the secretion of pro-inflammatory cytokines suggests that this pathway may facilitate the inflammatory milieu observed in UC [20,21].

Membrane trafficking (R-HSA-199991) is closely linked to vesicle-mediated transport and is critical for the proper localisation of membrane proteins, including receptors and transporters that regulate intestinal permeability. In UC, the dysregulation of membrane trafficking can lead to the aberrant expression of tight junction proteins, which are essential for maintaining epithelial barrier integrity [18,19]. This disruption can result in increased intestinal permeability, allowing luminal antigens to penetrate the epithelium and trigger an inflammatory response [20,21]. Moreover, the role of membrane trafficking in the internalisation of inflammatory mediators further underscores its significance in the pathogenesis of UC [19,22].

Formation of tubulin folding intermediates by CCT/TriC (R-HSA-389960) is essential for proper cytoskeletal dynamics, which are crucial for maintaining epithelial cell structure and function. In UC, the dysregulation of cytoskeletal proteins can lead to altered cell signalling and increased susceptibility to apoptosis, contributing to mucosal injury [18,23]. The CCT/TriC complex is involved in the folding of tubulin, and its dysfunction can impair microtubule organisation, affecting cell motility and the response to inflammatory stimuli [22,23]. This pathway’s involvement in the maintenance of epithelial integrity highlights its potential role in the chronic inflammation characteristic of long-lasting UC [20,21].

#### 3.2.2. Pathways Overlapping in Manifested and Pre-Diagnostic UC

Cooperation of prefoldin and TriC/CCT in actin and tubulin folding (R-HSA-389958) emphasises the importance of protein folding in cellular function. The interplay between prefoldin and the CCT complex is vital for the proper assembly of cytoskeletal components, which are crucial for maintaining epithelial barrier function [18,23]. In UC, disruptions in this pathway can lead to cytoskeletal instability, contributing to epithelial cell dysfunction and inflammation [20,21]. Furthermore, the role of this pathway in the regulation of immune cell trafficking suggests that it may also influence the inflammatory response in UC [19,22].

Chaperonin-mediated protein folding (R-HSA-390466) is another critical pathway that supports cellular homeostasis by ensuring the proper folding of proteins involved in inflammatory responses. In UC, the upregulation of chaperonins has been observed, indicating a cellular stress response to inflammation [18,20]. This pathway’s involvement in the stabilisation of proteins that regulate immune responses suggests that it plays a significant role in modulating the inflammatory milieu in UC [22,24]. Additionally, the dysregulation of chaperonin function can lead to the accumulation of misfolded proteins, further exacerbating inflammation [19,21].

VLDLR internalisation and degradation (R-HSA-8866427) are implicated in lipid metabolism and the regulation of inflammatory responses. The VLDLR pathway is involved in the internalisation of lipoproteins and their subsequent degradation, which can influence the availability of lipid mediators that modulate inflammation [18,20]. In UC, alterations in lipid metabolism and VLDLR function have been linked to the severity of inflammation, suggesting that this pathway may contribute to the pathogenesis of the disease [22,24]. Furthermore, the role of VLDLR in the regulation of immune cell function highlights its potential significance in the inflammatory processes associated with UC [19,21].

Viral infection pathways (R-HSA-9824446) have also been implicated in UC, particularly in the context of dysregulated immune responses. Viral infections can trigger inflammatory pathways that exacerbate UC symptoms, and the interplay between viral antigens and the host immune system can lead to chronic inflammation [18,20]. The association of viral infections with UC underscores the importance of understanding how viral pathways may interact with host immune responses to influence disease progression [22,24]. Additionally, the potential for viral infections to alter gut microbiota composition may further contribute to the pathogenesis of UC [19,21].

#### 3.2.3. Pathways Identified in Pre-Diagnostic UC

Eukaryotic translation elongation (R-HSA-156842) is crucial for protein synthesis and is often upregulated in response to cellular stress. In the context of UC, alterations in translation elongation can affect the production of proteins involved in inflammatory responses, potentially leading to an increased risk of disease onset [18,20]. The regulation of translation elongation may also influence the expression of cytokines and other mediators that contribute to the inflammatory process in UC [22,24]. Understanding the role of this pathway in the pre-diagnostic stages of UC may provide insights into early intervention strategies [19,21].

L13a-mediated translational silencing of ceruloplasmin expression (R-HSA-156827) highlights the importance of post-transcriptional regulation in the inflammatory response. The L13a protein is involved in the silencing of mRNAs that encode pro-inflammatory cytokines, and its dysregulation can lead to enhanced inflammation in UC [18,20]. This pathway’s role in modulating the expression of inflammatory mediators suggests that it may serve as a potential target for therapeutic intervention in pre-diagnostic UC [22,24]. Furthermore, the association of L13a with the resolution of inflammation underscores its significance in maintaining intestinal homeostasis [19,21].

GTP hydrolysis and joining of the 60S ribosomal subunit (R-HSA-72706) are essential for the assembly of ribosomes and the initiation of protein synthesis. In UC, alterations in ribosomal function can impact the synthesis of proteins involved in the inflammatory response, potentially influencing disease onset [18,20]. The regulation of ribosomal assembly and function may also play a role in the modulation of immune responses, highlighting the importance of this pathway in the context of UC [22,24]. Understanding how ribosomal pathways are altered in pre-diagnostic stages of UC may provide valuable insights into the mechanisms underlying disease development [19,21].

In conclusion, the Reactome pathways discussed above play significant roles in the pathogenesis of UC, influencing various aspects of inflammation, immune response, and epithelial integrity. Understanding these pathways provides a comprehensive view of the molecular mechanisms underlying UC and may inform future therapeutic strategies aimed at preventing or mitigating the disease.

### 3.3. Potential Relevance to UC Treatment Based on the Pathways

The pathogenesis of UC is multifactorial, involving immune dysregulation, epithelial barrier dysfunction, and alterations in gut microbiota. Recent insights into specific Reactome pathways have opened avenues for both existing and novel therapeutic strategies. Section 3 will explore the relevance of current therapeutics in UC treatment and highlight potential novel therapeutic options based on the identified pathways.

Current treatment modalities for UC primarily focus on reducing inflammation and managing symptoms. Conventional therapies include 5-aminosalicylic acid (5-ASA), corticosteroids, immunosuppressants, and biologics such as TNF-α inhibitors (e.g., infliximab) [25,26]. Although our study was not specifically designed to dissect treatment effects, applying an ANOVA model revealed significant antigenic reactivities associated with distinct treatment regimens. These findings suggest that serological testing for antibody reactivities could hold potential for patient stratification. Importantly, the biological relevance of the antibody profiles was supported by pathway analysis of the ANOVA-derived antigens, which revealed enrichment in pathways related to immune activation, signal transduction, and cellular responses to external stimuli. These pathways likely reflect a functional immune response signature characteristic of cells responding to infection, inflammation, or immune-modulating therapies such as Mesalazine, Cortisone, and TNFα-blockers. Their presence implies that our data may capture treatment-induced shifts in immune signalling and defence readiness.

While these therapies can be effective, they often come with significant side effects, including nausea, vomiting, and increased risk of infections and malignancies [27]. The reliance on anti-inflammatory drugs underscores the need for strategies that not only alleviate inflammation but also promote mucosal healing and restore epithelial integrity [27,28].

Recent studies have highlighted the importance of repairing the intestinal barrier as a therapeutic strategy. For instance, Guo et al. demonstrated that a human TFF2-Fc fusion protein promotes intestinal epithelial cell repair while inhibiting macrophage inflammation in a mouse model of DSS-induced colitis [27]. This approach aligns with the role of vesicle-mediated transport and membrane trafficking pathways in maintaining epithelial integrity, suggesting that therapies targeting these pathways could enhance mucosal healing in UC patients.

#### Novel Potential Therapeutic Approaches Inspired by Reactome Pathways

Targeting Epithelial Repair Mechanisms: Therapies that enhance vesicle-mediated transport and membrane trafficking could promote epithelial barrier integrity, aiding mucosal healing in UC patients. For instance, compounds that improve vesicle trafficking may help restore intestinal epithelial function and reduce inflammation [27,28].

Modulating Immune Responses: Enhancing the function of pathways involved in protein folding, such as prefoldin and TriC/CCT, could help regulate immune responses. For example, compounds like Bergenin have shown potential in modulating gut microbiota and reducing inflammation, indicating that microbiota-targeted therapies may be beneficial [29]. This is in line with utilising gut microbiota modulation and restoring a healthy gut microbiota balance, which could alleviate UC symptoms. Genetic and genomic approaches may identify novel drug targets for microbiome-based therapies, enhancing treatment efficacy [28].

The development of innovative drug delivery systems, such as nanoparticles for targeted drug delivery, has the potential to enhance the efficacy of existing therapies while minimising side effects. Targeted nanoparticles have shown promise in preventing mucosal damage and alleviating inflammation in UC [30]. Additionally, investigating the role of autophagy in inflammation suggests that boosting mTOR-dependent autophagy may help regulate intestinal inflammation, offering a potential novel strategy for managing refractory UC [31].

In summary, leveraging insights from Reactome pathways offers promising avenues for developing novel therapeutic strategies in UC, focusing on epithelial repair, immune modulation, microbiota restoration, and innovative delivery systems.

### 3.4. Autoantibodies in UC

Autoantibodies are proteins produced by the immune system that mistakenly target the body’s own tissues. In the context of UC, specific autoantibodies have been identified that correlate with disease activity and severity. For instance, in our previous work, we discuss the presence of autoantibodies as diagnostic markers and their potential role in driving inflammation in UC, particularly emphasising the IL-10 and IL-23 pathways that modulate immune responses and autoantibody activity [9]. Furthermore, the study by Uzzan et al. indicates that UC is characterised by a plasmablast-skewed humoral response, which is associated with disease activity, suggesting that the expansion of certain B cell populations may contribute to the production of autoantibodies in UC patients [32].

The immune dysregulation observed in UC is further supported by findings from He et al., who highlight that dysregulated immune responses, particularly involving neutrophils, play a crucial role in the development of UC. Excessive recruitment of activated neutrophils leads to mucosal injury and inflammation, which may be exacerbated by the presence of autoantibodies [33]. Additionally, the work of Giannos et al. points to specific autoantibodies, such as those against osteopenia, which are significantly elevated in UC patients compared to those with Crohn’s disease, indicating a unique immune profile associated with UC [34].

Moreover, the role of oxidative stress in UC pathogenesis cannot be overlooked. Ipek et al. discuss how oxidative stress, driven by reactive oxygen species, contributes to tissue injury in UC, which may interact with autoantibody production and activity [35]. This interplay between oxidative stress and autoimmunity is critical, as it may influence the severity of inflammation and the overall disease course.

In summary, the reactivity of autoantibodies in UC is a multifaceted phenomenon that involves complex interactions between immune cells, cytokines, and oxidative stress. The identification of specific autoantibodies, as shown by this study, not only enhances our understanding of the disease mechanisms but also opens avenues for novel diagnostic and therapeutic strategies aimed at modulating the immune response in UC patients.

## 4. Materials and Methods

### 4.1. Samples

For autoantibody profiling of manifested UC, blood samples were collected from 49 UC patients and 23 non-affected individuals. For basic demographics, see Table 1. Written consent was given by all donors, and the study was approved by the Institutional Review Board (IRB) of the Medical Faculty at the University of Munich (120–15). The clinical activity score was assessed by using the simple clinical colitis activity index (SCCAI) [36].

Similarly, antibody profiling was performed using a pre-diagnostic cohort obtained from the biobank of the Bavarian Red Cross (BRK). A total of 66 plasma samples were identified from the BRK biobank, including 33 samples from 11 patients later diagnosed with UC, each with three samples collected at different time points prior to diagnosis. These were compared with 33 healthy controls, matched by gender, as well as by the time and centre of blood donation. No additional clinical data were available, but these “non-diseased controls” met the general eligibility criteria for blood donation. The sample characteristics of the pre-diagnostic UC patients are summarised in Table 4.

### 4.2. Isolation of Immunoglobulin

IgG was purified from serum samples (UC patients: *n* = 49; non-UC controls: *n* = 23) and plasma samples (pre-diagnostic samples from BRK) using the Melon Gel™ IgG Spin Purification Kit (Thermo Fisher Scientific™, Vienna, Austria, Cat. No. 45,206). Patient serum or plasma was diluted 1:10 with the kit’s buffer, and purification was carried out following the manufacturer’s protocol. IgG concentrations were measured in duplicate using A280 spectrophotometry (Epoch Take3 system, BioTek Instruments Inc., Winooski, VT, USA) and adjusted to 0.4 mg/mL with the kit-provided buffer. The prepared IgG was then stored at −20 °C until further processing for protein microarray analysis. For all samples, IgG concentration approximation was based on the quantification of purified IgG obtained using the Melon™ Gel IgG Spin Purification Kit (Thermo Fisher Scientific, Waltham, MA, USA). As all samples were processed uniformly—using the same starting volume, elution volume, and identical protocol—this approach enabled a consistent relative comparison. Potential subclass-specific interactions with the purification matrix were considered negligible under these conditions. While the manufacturer states that IgG recovery exceeds 90% with >80% purity, it is acknowledged that recovery may vary slightly between samples and is not necessarily complete. Thus, although this method does not guarantee absolute quantification, it provides an acceptable and standardised approximation of IgG concentration across the dataset, particularly given that protein levels were determined via absorbance at 280 nm.

### 4.3. Protein Microarray Processing and Microarray Data Analysis

Protein array processing, including image acquisition, data extraction, and data analysis, was conducted as described in detail by Jodeleit [9] and Milchram [37]. 

In short:

Standardised concentrations of 0.2 mg IgG/mL, applying 450 µL per microarray, were used for antibody profiling. Patient’s IgG bound onto antigens presented on microarrays was detected by a fluorescently labelled anti-human IgG detection antibody.

Protein microarray analysis was conducted using AIT’s 16k protein microarray presenting proteins recombinantly expressed in *E. coli* from 15,286 human cDNA expression clones. These so-called UNIPEX clones represent full-length cDNA clones, which were selected from expression clones derived from human foetal brain (*n* = 7383) as well as T-cells, lung, and colon tissue (*n* = 7903; personal information provided by RZPD). These UNIPEX clones represent 6369 different human proteins (each protein presented by 2 or more different clones), for which 5449 have been annotated with a gene symbol, when the cDNA sequence perfectly matched to database entries.

Recombinant proteins were produced, purified, and spotted on SU8 epoxy-dip-coated slides using a NanoPrint™ LM210 device (Arrayit Corp, San Francisco, CA, USA). Quality control included anti-His Tag antibody staining and reproducibility checks. The microarrays were blocked, hybridised with 0.2 mg/mL IgG in PBST with skimmed milk, and incubated for four hours at room temperature. Post-incubation, the slides were washed and probed with Alexa Fluor 647-labelled anti-human IgG (Thermo Fisher Scientific™, Vienna, Austria, Cat. No. A-21445) for detection. Final washing and drying ensured slides were prepared for image acquisition.

Slides were scanned using a Tecan LS 200 Microarray scanner (Tecan Austria GmbH, Grödig, Austria), and raw TIFF images were processed with GenePix Pro 6.0 (Molecular Devices LLC, San Jose, CA, USA) to extract data. Preprocessing involved log2 transformation: background-corrected median relative fluorescent intensities (RFI) from microarray images were log2 transformed. The median normalised Log2-RFI values were then bio-statistically evaluated with BRB Array Tools Version 4.5.0 (National Cancer Institute, Bethesda, MD, USA; https://brb.nci.nih.gov/BRB-ArrayTools.html, accessed on 24 April 2025).

### 4.4. Pathway Analysis of Antigens Derived from Protein Microarray Data Analysis

Pathway enrichment analysis was conducted using WebGestalt (WEB-based Gene SeT AnaLysis Toolkit, https://www.webgestalt.org/, accessed on 31 January 2025) to identify significantly enriched biological pathways. The over-representation analysis (ORA) method was applied, utilising the Reactome pathway database as the reference.

Input data and mapping: The input list from the class comparison analysis was used for analyses and as background gene set; the 6125 single gene symbol names of proteins present on AIT’s 16k arrays, mapped to 6084 Entrez Gene IDs, with 3974 IDs annotated to selected functional categories were used.

The default enrichment analysis parameters have been used as follows: minimum number of IDs per category: 5; maximum number of IDs per category: 2000; statistical significance of enrichment was determined using Fisher’s Exact Test, followed by FDR correction using the Benjamini–Hochberg procedure. The resulting significantly enriched pathways (FDR < 0.05) were further used for discussion and to interpret biological implications.

## 5. Conclusions

This study underscores the potential of autoantibody profiling as a valuable tool for the early detection and mechanistic understanding of UC. By analysing autoantibody responses in both severe UC patients and pre-diagnostic cohorts, we have identified DIRAGs and disease activity-related antigens that correlate with disease progression. The significant overlap observed across distinct sample types—serum and plasma, as well as manifested and pre-diagnostic UC—reinforces the robustness and reliability of our findings.

Our results align with emerging evidence that autoantibodies may precede clinical UC onset by several years, providing a window for early diagnosis and intervention. Furthermore, the integration of Reactome pathway analysis has allowed us to map these immune signatures to underlying molecular pathology, offering an additional omics layer for UC research. By bridging diagnostic biomarker discovery with pathophysiological insights, this study not only enhances our understanding of UC pathogenesis but also paves the way for novel therapeutic strategies and personalised patient care.

## 6. Patents

Biomarkers derived from this study have been claimed in the patent application EP3631456A1–in this paper, we focus on the link to the known published literature and pathway analyses.

## Figures and Tables

**Figure 1 ijms-26-04086-f001:**
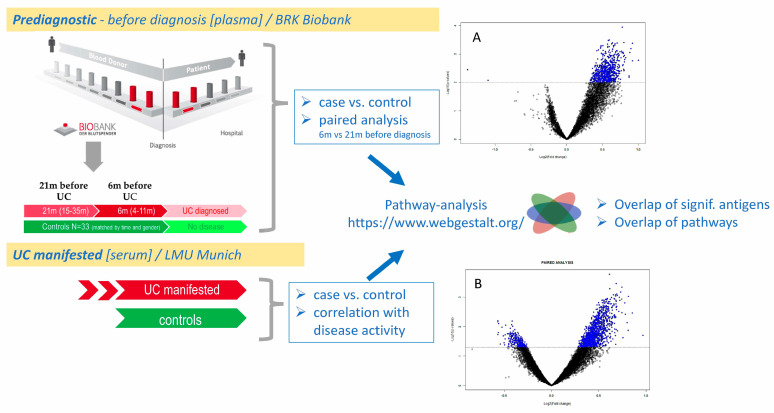
This overview illustrates the overall study concept, comparing autoantibody profiles across two ulcerative colitis (UC) cohorts: (1) a manifested UC cohort (serum samples from LMU Munich) and (2) a pre-diagnostic UC cohort (plasma samples from the Bavarian Red Cross biobank). In the pre-diagnostic cohort, samples were collected longitudinally at two time points: median 21 (15–35) months and median 6 (4–11) months prior to clinical diagnosis. Autoantibody reactivity was analysed using a high-content 16k protein microarray platform. Volcano plots (blue dots indicate significant features) in the figure are according to our previous paper [9]. Key comparisons included: case vs. control analyses in both cohorts; paired longitudinal analysis within the pre-diagnostic group; correlation with simple clinical colitis activity index (SCCAI) score in manifested UC, as well as pathway analyses. (**A**) Autoantibodies as biomarkers to discriminate between UC and non-UC donors. Increased autoantibody reactivity in UC patients is displayed as a volcano plot. Purified serum IgG of UC (49) and non-UC (23) patients were subjected to protein-microarray analysis of UC vs. non-UC. Antigenic reactivity in UC vs. non-UC is presented. Significant DIRAGs are given in blue (*n* = 691; *p* < 0.01; x-axis shows the “−log10 *p* value”; y-axis shows the log2—FC between classes. See Appendix A). (**B**) Volcano plot derived from paired analysis of T3 vs. T-early samples. The majority of 1201 features have higher reactivity in samples at T3 (closest to clinical UC presentation); vice versa, only 242 of the total 1371 significantly reactive antigens have higher reactivity at T-early. Although the threshold for significance was set to a moderate *p* < 0.05 value, and thus false positives might be relatively high, the overwhelming number of antigens showing a higher reactivity at time close to diagnosis is in line with the findings described in (**A**), where severe UC (undergoing therapy) vs. controls showed a similar pattern, indicating breach of tolerance and high autoantibody reactivity vs. self-antigens.

**Table 1 ijms-26-04086-t001:** Patient samples, duration of disease, and therapy of patients (treatment information is provided in Appendix A).

	UC *n* = 49	Active UC *n* = 19	Non-UC *n* = 23
**Age (years)**			
Mean (SD)	38.5 (15.6)	36.13	36.7 (15.9)
Range	24–74	19–71	21–59
**Gender (% male)**	46	42	42
**Duration of UC (years)**			
Mean (SD)	11.6 (9.53)	11.46 (10.97)	
Range	1–40	1–39	
**SCCAI**			
Mean (SD)	3.04 (2.79)	6.42 (2.19)	
Range	0–13	5–13	
**Treatment (current)**			
TNFα-blocker	20	6	
Glucocorticoids	13	8	
Mesalazine	26	9	
Immuno-suppressive	6	3	
No	10	3	
**Atopic Dermatitis**			3

**Table 2 ijms-26-04086-t002:** DIRAGS in UC vs. control—top 20 sorted by FC derived from 691 DIRAGS higher reactive in manifested UC vs. controls (see Appendix A).

Parametric *p*-Value	FDR	Geom Mean of Intensities in Class 1 (UC)	Geom Mean of Intensities in Class 2 (Contr)	Fold Change	UniqueID	Name (Updated)
0.0016529	0.0795	1282.1	640.82	2	RZPDp9027O0413Q	DMPK
0.0004156	0.0795	1788.2	947.7	1.89	RZPDp9027K1313Q	HES1
0.001753	0.0795	1818.41	962.46	1.89	RZPDp9027F128Q	GGA1
0.0006475	0.0795	1283.67	690.63	1.86	RZPDp9027A1813Q	RPS29
0.0008848	0.0795	1485.46	805.73	1.84	RZPDp9027C1612Q	CD99
0.005878	0.0796	937.25	513.29	1.83	RZPDp9027B0418Q	TRIM27
0.0065022	0.0825	2056.95	1124.54	1.83	RZPDp9027B186Q	MCM3AP
0.0013648	0.0795	2432.38	1374.53	1.77	RZPDp9027A1412Q	POGZ
0.0015698	0.0795	2069.3	1180.18	1.75	RZPDp9027H0515Q	SLC16A8
0.0018793	0.0795	856.75	490.21	1.75	RZPDp9027C1713Q	FLAD1
0.0003951	0.0795	354.27	203.84	1.74	RZPDp9027D1013Q	NBPF9
0.001114	0.0795	467.47	267.9	1.74	RZPDp9027J0210Q	SCAP
0.0036918	0.0795	2389.08	1371.05	1.74	RZPDp9027H1111Q	CD99
0.00383	0.0795	541.62	311.4	1.74	RZPDp9027C138Q	IP6K2
0.0009492	0.0795	7994.23	4619.48	1.73	RZPDp9027J2111Q	POLR3H
0.0012119	0.0795	1889.77	1089.48	1.73	RZPDp9027D2216Q	RPS17
0.0023996	0.0795	810.49	469.71	1.73	RZPDp9027K2414Q	UBXN4
0.0034252	0.0795	512.57	295.68	1.73	RZPDp9027B1813Q	YES1
0.0008638	0.0795	1072.27	622.45	1.72	RZPDp9027A1312Q	AP2S1
0.0025636	0.0795	1071.97	624.53	1.72	RZPDp9027L1713Q	SEPTIN7P14

**Table 3 ijms-26-04086-t003:** Correlation analysis of protein array data with SCCAI in UC patients (*n* = 43). (A) Top antigens ordered by Pearson correlation coefficients and *p*-values derived from “Median normalised data” and (B) same as in (A) but derived from unnormalised data. Proteins that were found to be significantly correlated with disease activity in both median-normalised and unnormalised datasets are marked in the tables. These shared proteins are CLTA, ZWINT, and TUFM.

(A) Median Normalised	Correlation Coefficient	Parametric *p*-Value	Name
1	−0.444	0.0013822	PODXL2
2	−0.444	0.0013919	FBLL1
3	−0.437	0.00171	PHF19
4	−0.416	0.0029337	PRPF3
**5**	**−0.414**	**0.0031113**	**CLTA**
6	−0.413	0.0032041	TMEM44
7	−0.406	0.0037712	CHAMP1
8	−0.404	0.0039976	TTLL7
9	−0.396	0.0048621	CLEC11A
10	−0.393	0.0051929	PIPSL
11	−0.391	0.0054845	ATXN10
12	−0.391	0.0055268	KNDC1
**13**	**−0.39**	**0.00566**	**ZWINT**
**14**	**−0.386**	**0.0062099**	**TUFM**
15	−0.379	0.0071869	USP34
16	−0.378	0.0074131	SUPV3L1
17	−0.377	0.0076329	A2M
18	−0.376	0.0078365	PAICS
19	0.374	0.0080463	PDIA4
20	−0.373	0.0082321	CCT8
21	−0.371	0.008659	XRCC6
22	−0.371	0.0087122	ROBO3
23	−0.368	0.0092002	OLA1
24	−0.368	0.0093	BSG
25	−0.367	0.0094562	PRMT2
26	−0.365	0.0099094	MEIS3
**(B) Un-** **Normalised**	**Correlation** **Coefficient**	**Parametric** ***p*-Value**	**Name**
1	−0.57	5.40 × 10^−5^	DCAF13
2	−0.53	9.01 × 10^−5^	ACO2
3	−0.528	9.62 × 10^−5^	C17orf70
4	−0.524	0.0001106	TPP2
5	−0.522	0.0001681	RPL7A
6	−0.51	0.0001831	VARS2
7	−0.506	0.0002098	RAB5C
8	−0.502	0.0002377	DHX30
9	−0.5	0.0002513	PTPN6
10	−0.487	0.0003836	MINA
11	−0.477	0.0005248	CNTD1
12	−0.476	0.0005513	MCM6
**13**	**−0.475**	**0.0005583**	**ZWINT**
14	−0.472	0.0006137	RSRP1
15	−0.472	0.0006261	GOLGA2
**16**	**−0.471**	**0.0006396**	**CLTA**
**17**	**−0.464**	**0.0007813**	**TUFM**
18	−0.464	0.0007823	CCDC94
19	−0.46	0.0008893	HSF1
20	−0.458	0.0009435	CCT2
21	−0.456	0.0009975	FLNA

**Table 4 ijms-26-04086-t004:** UC-pre-diagnostic sample characteristics. Samples were obtained from the blood bank of the BRK—biobanked prior clinical UC diagnosis. * IgG concentration measures upon MelonGel purification; ** increase in IgG concentration in T3 was evaluated with respect to the samples taken >12 m before diagnosis, i.e., 9–24 months (median delta T3–Tx of 13 months; these were collected 15–35 [median 21] months before UC diagnosis) before T3 (compared samples and their IgG concentrations are indicated by bold numbers in the time and IgG concentration columns, respectively); cases 4 and 6 were excluded when evaluating “IgG increase” because for case 4 all samples were taken <7 months prior UC clinical diagnosis, and for case 6 all samples were collected >50 months prior UC clinical diagnosis. *** case 2 and 3 failed classification, see chapter paired analysis.

UC Individuals	Year of Diagnosis	Age at Diagnosis (y)	Gender	Blood Group (AB0)	Time (month) Sample Taken Prior Diagnosis			IgG Concentration * [mg/mL plasma]			IgG Increase (%) **	Delta T3–Tx (m) **
					T1	T2	T3	IgG T1	IgG T2	IgG T3		
case 1	2008	45	M	0	22	**20**	**4**	7.95	**8.70**	**9.24**	6%	16
case 2 ***	2004	35	W	A	**19**	16	**4**	**7.17**	7.47	**8.78**	22%	15
case 3 ***	2005	61	M	0	**15**	10	**6**	**7.01**	6.37	**6.59**	–6%	9
case 4	2004	45	M	A	7	4	**1**	7.32	7.59	**8.34**		
case 5	2005	46	W	A	**17**	12	**4**	**9.64**	8.94	**7.11**	–26%	13
case 6	2011	35	M	0	55	53	50	7.07	7.49	**7.31**		
case 7	2009	42	W	A	34	**24**	**11**	7.17	**5.33**	**8.18**	53%	13
case 8	2009	31	W	A	25	**22**	**10**	8.29	**5.87**	**6.27**	7%	12
case 9	2010	43	M	0	65	**22**	**3**	8.71	**5.91**	**7.60**	29%	19
case 10	2012	49	W	B	**21**	14	**11**	**10.70**	5.23	**4.33**	–60%	10
case 11	2012	44	M	0	**35**	20	**11**	**5.47**	5.48	**7.03**	29%	24
median		44					6			**7.31**	6.8%	13

**Table 5 ijms-26-04086-t005:** The top 20 antigenic proteins ranked by fold change (FC) derived from paired analysis of pre-diagnostic sample analysis—the full list is provided in Appendix A. FC of geometric mean intensities of classes as indicated in the header; protein names highlighted in bold (SRSF9, DISP3, USP11, HNRNPA2B1, FAU) indicate proteins presented from 2 different expression clones, where both were found significant.

Protein/Gene Symbol	Parametric *p*-Value	Fold Change (T3 vs. T-Early)
DCAF5	0.0034474	1.81
**SRSF9**	0.0009296	1.76
LAMC1	0.0131338	1.75
SULT1A3	0.0072182	1.74
EVL	0.0038099	1.72
CKAP5	0.0015792	1.71
TINF2	0.0235447	1.69
**DISP3**	0.0004463	1.68
NAP1L4	0.0033596	1.68
PTPRE	0.001167	1.67
EIF2S3	0.0006555	1.66
TIAL1	0.0012767	1.65
FAM13A	0.0010115	1.64
**USP11**	0.0030261	1.64
FMNL2	0.0008524	1.62
**HNRNPA2B1**	0.0139107	1.62
PPID	0.0003352	1.61
MAGED2	0.0007698	1.61
SUGP2	0.001397	1.61
**FAU**	0.0020518	1.61

**Table 6 ijms-26-04086-t006:** Top features sorted by FC derived from paired analysis of 9 pairs of patients (T3 vs. T-early; FC ≥ 1.5; univariate misclassification rate below 0.001).

ANTIGENS	Parametric *p*-Value	t-Value	Fold Change
DCAF5	0.0034474	3.397	1.81
PPID	0.0003352	4.476	1.61
GBAS	0.0008214	4.059	1.6
LRP5	0.001681	3.728	1.58
PKM	0.000819	4.06	1.57
WNK2	0.001018	3.96	1.55
POGLUT1	0.0011713	3.895	1.54
AAAS	0.0001634	4.815	1.53
LMO4	0.0025234	3.541	1.52
AP2S1	0.0015832	3.756	1.51
SLC22A17	0.0030934	3.447	1.5

**Table 7 ijms-26-04086-t007:** Distinguishing UC before clinical manifestation vs. samples 2 years before. Top features are sorted by FC derived from paired analysis of 7 pairs of patients (T3 vs. T-early; *p* < 5 × 10^−4^ significance level and univariate misclassification rate below 1 × 10^−6^).

ANTIGENS	Parametric *p*-Value	t-Value	Fold Change
EVL	0.0002057	4.818	2.1
HNRNPA2B1	0.0000997	5.184	2.1
CD58	0.0000735	5.34	2.06
EEF1A1	0.0004424	4.439	2.01
DRG1	0.0000945	5.211	1.98
DHX8	0.0001773	4.893	1.95
GNAI2	0.0001624	4.936	1.95
ALKBH2	0.0001302	5.048	1.95
BCAR1	0.0000853	5.264	1.93
TIAL1	0.0001786	4.889	1.92
FADS1	0.0000979	5.193	1.92
EIF2S3	0.0001117	5.126	1.91
PTCHD2	0.0000894	5.24	1.91
LAMTOR1	0.000414	4.471	1.89
RNF10	0.0003557	4.546	1.89
FAM209B	0.0003529	4.55	1.89
CDCA4	0.0002964	4.636	1.89
FAM13A	0.000187	4.866	1.89
SUGP2	0.0000849	5.266	1.89
CPNE1	0.0003801	4.513	1.88
VWF	0.0003278	4.586	1.88
GOT1	0.0000907	5.232	1.88
ZNF84	0.0002809	4.663	1.87
RUNDC3A	0.0001665	4.924	1.87
FMNL2	0.0001064	5.151	1.87
RNASEK	0.0001831	4.876	1.86
C17orf62	0.0001416	5.006	1.84
FAM131A	0.0004892	4.389	1.83
MINK1	0.000285	4.656	1.83
SH3BGRL3	0.0001401	5.011	1.83
SIRT6	0.0002133	4.8	1.81
CKB	0.0002853	4.655	1.8
PLXNB2	0.0004156	4.47	1.78
CCT3	0.0003019	4.627	1.78
UBE2L3	0.0002807	4.663	1.78
USP28	0.0001105	5.132	1.77
TTC19	0.0001416	5.006	1.76
EIF4A2	0.0004477	4.433	1.75
FTSJD2	0.0003739	4.522	1.75
R3HDM1	0.0004045	4.483	1.74
PPID	0.0004417	4.44	1.73
WNK2	0.000428	4.455	1.73
HSP90AB1	0.0001805	4.883	1.73
PDHA1	0.0004512	4.429	1.72
KIAA1731	0.0003517	4.552	1.72
RPL36A	0.0002	4.832	1.72
DST	0.0001255	5.067	1.71
CAPZA2	0.0001205	5.087	1.71
EIF3G	0.0004173	4.468	1.7
SRSF2	0.0003381	4.571	1.7
DYNC1I2	0.0003514	4.552	1.68
NARS2	0.0002444	4.732	1.67
ZCCHC11	0.0004672	4.412	1.65
GTF2I	0.0004434	4.438	1.65
MSL1	0.0004614	4.418	1.61
AAAS	0.0004851	4.394	1.6

**Table 8 ijms-26-04086-t008:** Pathways derived from Webgestalt analysis (A) of significantly enriched antigens in the “severe UC case/control” setting (using the list of 691 antigens) and (B) the “pre-diagnostic UC samples” study (“T-early vs. T3 at diagnosis”; using the list of 1371 antigens). Gene sets marked bold are found overlapping between both pathway lists A and B.

(A)	Gene Set	Description	Size	Expect	Ratio	*p* Value	FDR
1	R-HSA-5653656	Vesicle-mediated transport	350	38.576	1.7109	3.6111 × 10^−6^	0.0045419
2	R-HSA-199991	Membrane Trafficking	327	36.041	1.7203	6.1879 × 10^−6^	0.0045419
3	R-HSA-446203	Asparagine N-linked glycosylation	134	14.769	2.1667	0.000012865	0.0062954
4	R-HSA-2132295	MHC class II antigen presentation	75	8.2662	2.5405	0.000034295	0.0096775
5	R-HSA-6807878	COPI-mediated anterograde transport	59	6.5028	2.7681	0.00003565	0.0096775
**6**	**R-HSA-389960**	**Formation of tubulin folding intermediates by CCT/TriC**	**18**	**1.9839**	**4.5365**	**0.000043264**	**0.0096775**
7	R-HSA-199977	ER to Golgi Anterograde Transport	82	9.0377	2.4342	0.000046146	0.0096775
8	R-HSA-389957	Prefoldin mediated transfer of substrate to CCT/TriC	23	2.535	3.9448	0.000072494	0.013303
9	R-HSA-390450	Folding of actin by CCT/TriC	9	0.99195	6.0487	0.00010891	0.014974
10	R-HSA-437239	Recycling pathway of L1	33	3.6371	3.2993	0.00011214	0.014974
**11**	**R-HSA-389958**	**Cooperation of Prefoldin and TriC/CCT in actin and tubulin folding**	**24**	**2.6452**	**3.7804**	**0.0001122**	**0.014974**
12	R-HSA-390466	Chaperonin-mediated protein folding	44	4.8495	2.8869	0.00016066	0.018747
**13**	**R-HSA-9646399**	**Aggrephagy**	**25**	**2.7554**	**3.6292**	**0.00016885**	**0.018747**
14	R-HSA-8866427	VLDLR internalisation and degradation	13	1.4328	4.8855	0.00017879	0.018747
15	R-HSA-948021	Transport to the Golgi and subsequent modification	90	9.9195	2.2179	0.00020769	0.020326
**16**	**R-HSA-5663205**	**Infectious disease**	**527**	**58.084**	**1.429**	**0.00023271**	**0.021332**
17	R-HSA-177504	Retrograde neurotrophin signalling	10	1.1022	5.4438	0.00024703	0.021332
18	R-HSA-390471	Association of TriC/CCT with target proteins during biosynthesis	22	2.4248	3.7117	0.00029508	0.024066
19	R-HSA-901042	Calnexin/calreticulin cycle	14	1.543	4.5365	0.00032389	0.02411
20	R-HSA-9734009	Defective Intrinsic Pathway for Apoptosis	18	1.9839	4.0325	0.00032847	0.02411
21	R-HSA-391251	Protein folding	48	5.2904	2.6463	0.00044476	0.031091
22	R-HSA-9663891	Selective autophagy	44	4.8495	2.6807	0.0006185	0.041271
23	R-HSA-373760	L1CAM interactions	61	6.7232	2.3798	0.00066087	0.042181
**24**	**R-HSA-9824446**	**Viral Infection Pathways**	**443**	**48.826**	**1.4337**	**0.00071004**	**0.043431**
**(B)**	**Gene Set**	**Description**	**Size**	**Expect**	**Ratio**	***p* Value**	**FDR**
1	R-HSA-156842	Eukaryotic Translation Elongation	81	16.612	2.1069	2.4876 × 10^−6^	0.001445
2	R-HSA-156827	L13a-mediated translational silencing of Ceruloplasmin expression	99	20.303	1.9701	3.7689 × 10^−6^	0.001445
3	R-HSA-72706	GTP hydrolysis and joining of the 60S ribosomal subunit	99	20.303	1.9701	3.7689 × 10^−6^	0.001445
4	R-HSA-72689	Formation of a pool of free 40S subunits	89	18.252	2.0271	3.9373 × 10^−6^	0.001445
5	R-HSA-72613	Eukaryotic Translation Initiation	104	21.329	1.9223	5.9519 × 10^−6^	0.0014562
6	R-HSA-72737	Cap-dependent Translation Initiation	104	21.329	1.9223	5.9519 × 10^−6^	0.0014562
7	R-HSA-2262752	Cellular responses to stress	445	91.262	1.3916	0.00001093	0.0022921
8	R-HSA-8953897	Cellular responses to stimuli	447	91.672	1.3854	0.000013969	0.0025634
9	R-HSA-156902	Peptide chain elongation	77	15.791	2.0264	0.000017954	0.0029286
10	R-HSA-192823	Viral mRNA Translation	75	15.381	2.0154	0.000027611	0.0040532
11	R-HSA-3371497	HSP90 chaperone cycle for steroid hormone receptors (SHR) in the presence of ligand	38	7.7932	2.438	0.000045552	0.0059558
12	R-HSA-72764	Eukaryotic Translation Termination	77	15.791	1.9631	0.000050981	0.0059558
13	R-HSA-9633012	Response of EIF2AK4 (GCN2) to amino acid deficiency	84	17.227	1.9156	0.000052742	0.0059558
14	R-HSA-975956	Nonsense Mediated Decay (NMD) independent of the Exon Junction Complex (EJC)	81	16.612	1.9264	0.000060108	0.0063027
15	R-HSA-2408522	Selenoamino acid metabolism	93	19.073	1.8351	0.000089636	0.0083464
16	R-HSA-2408557	Selenocysteine synthesis	79	16.202	1.9134	0.000090969	0.0083464
17	R-HSA-71291	Metabolism of amino acids and derivatives	199	40.812	1.5437	0.000096924	0.0083697
18	R-HSA-9711097	Cellular response to starvation	109	22.354	1.7447	0.00013029	0.010626
19	R-HSA-168273	Influenza Viral RNA Transcription and Replication	110	22.559	1.7288	0.00016304	0.011661
20	R-HSA-927802	Nonsense-Mediated Decay (NMD)	92	18.868	1.802	0.00017081	0.011661
21	R-HSA-975957	Nonsense Mediated Decay (NMD) enhanced by the Exon Junction Complex (EJC)	92	18.868	1.802	0.00017081	0.011661
22	R-HSA-72702	Ribosomal scanning and start codon recognition	54	11.074	2.0768	0.00017476	0.011661
23	R-HSA-72695	Formation of the ternary complex, and subsequently, the 43S complex	48	9.844	2.1333	0.00021176	0.013516
24	R-HSA-72649	Translation initiation complex formation	55	11.28	2.0391	0.00024325	0.014879
**25**	**R-HSA-5663205**	**Infectious disease**	**527**	**108.08**	**1.2861**	**0.00029014**	**0.016485**
26	R-HSA-1799339	SRP-dependent cotranslational protein targeting to membrane	87	17.842	1.7935	0.00029196	0.016485
27	R-HSA-72662	Activation of the mRNA upon binding of the cap-binding complex and eIFs, and subsequent binding to 43S	56	11.485	2.0027	0.00033432	0.018177
28	R-HSA-8876725	Protein methylation	9	1.8457	3.7925	0.00036366	0.019066
29	R-HSA-9612973	Autophagy	85	17.432	1.7783	0.00043017	0.020496
30	R-HSA-168255	Influenza Infection	126	25.84	1.6254	0.00043201	0.020496
31	R-HSA-422475	Axon guidance	316	64.806	1.3733	0.00043281	0.020496
**32**	**R-HSA-9646399**	**Aggrephagy**	**25**	**5.1271**	**2.5356**	**0.00045665**	**0.020949**
33	R-HSA-72766	Translation	195	39.991	1.4753	0.00061669	0.027433
34	R-HSA-9675108	Nervous system development	328	67.267	1.3528	0.00064718	0.027943
35	R-HSA-9010553	Regulation of expression of SLITs and ROBOs	133	27.276	1.5765	0.00076776	0.032131
36	R-HSA-8953854	Metabolism of RNA	481	98.645	1.2773	0.00078795	0.032131
**37**	**R-HSA-9824446**	**Viral Infection Pathways**	**443**	**90.852**	**1.2878**	**0.00088965**	**0.035298**
38	R-HSA-9735869	SARS-CoV-1 modulates host translation machinery	33	6.7677	2.2164	0.0010492	0.040532
**39**	**R-HSA-389960**	**Formation of tubulin folding intermediates by CCT/TriC**	**18**	**3.6915**	**2.7089**	**0.001089**	**0.04099**
**40**	**R-HSA-389958**	**Cooperation of Prefoldin and TriC/CCT in actin and tubulin folding**	**24**	**4.922**	**2.438**	**0.0011914**	**0.04372**

## Data Availability

Results data are provided in the Appendix A; protein array raw data will be provided upon request.

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
