# Peer review of "Autoantibody Profiling in Ulcerative Colitis: Identification of Early Immune Signatures and Disease-Associated Antigens for Improved Diagnosis and Monitoring"

_ijms, 2025, doi:10.3390/ijms26094086_

Round 1
Reviewer 1 Report
Comments and Suggestions for Authors
Results presented in ijms-3525991 could be of great importance as they could provide a basis for serology-based early UC diagnostics and following up of UC course. Manuscript presents results on autoreactivity of IgG from serum / plasma collected from UC patient during pre-diagnostic period as well as upon confirmation of UC. Trough comparison with reactivity of IgG isolated from serum of non-UC donors, several self-antigens are identified as specific targets of IgG in UC.
Reviewing of the manuscript was hampered significantly by improper citation of Tables and Figures throughout the text (“Error! Reference source not found.” at multiple position within a text). In addition, some parts of the Manuscript seem to be in a draft form (for example Section 2.4.5).
Although this is an interesting study, there are some points that need to be considered:
- Lines 107-110, 197-199, 207-209: Concentrations of IgG were determined after purification. Actually, they represent a yield of a purification process. Although they may mirror IgG concentration in the blood, they could not be directly extrapolated to IgG concentration in the blood. This part should be revised.
- It would be necessary to provide more details on non-UC patients (Diagnosis other than atopic dermatitis? Are they subjected to some therapeutic regimen?) and eventually take them into discussion. Details on controls for prediagnosed samples should be provided as well.
- I find that grouping of samples collected during prediagnostic period is not appropriate for making discrimination between period close to diagnosis (T3) and so called early (T1 and T2) period. For example (Table 4), sample taken 7 months prior diagnosis from patient 4 is assigned to T1 (early) while sample taken 50 months prior diagnosis from patient 6 is assigned to T3 (close to diagnosis). I suggest grouping of samples based on time frame between their collection and getting of UC diagnosis, rather than on simple chronology of their collection.
- Whether plots at Figure 1 and Figure 2 are already published in Jodeleit H et al, 2020 (https://doi.org/10.1371/journal.pone.0228615, Figure 1)?
- What is the difference between CAS and SCCAI? It seems that they refer to the same. Hence, one term should be chosen and used consistently trough the manuscript.
- Study evaluated self-antigens targeted specifically by IgG from UC patients. These antigens might eventually imply on the role of some genes, but genes are not directly identified in this study. In line, some parts of the Discussion should be revised.
- Line 493 – “genes” should be deleted; line 721 – IgG is missing after anti-human…; line 723 – please check “proteins recombinantly expressed in E.coli from human 15286 cDNA expression clones…”
Manuscript should be checked carefully for typo errors.
Author Response
Dear Reviewer,
Thank you very much for your thoughtful and constructive comments. We have carefully considered all your points and revised the manuscript accordingly - We have compiled all our edits in the word-flie attached.
We sincerely appreciate your suggestions and the time you invested in reviewing our work.
With kind regards,
On behalf of all authors, Andreas Weinhäusel

Reviewer 2 Report
Comments and Suggestions for Authors
In this manuscript, ,, Autoantibody Profiling in Ulcerative Colitis: Identification of Early Immune Signatures and Disease-Associated Antigens for Improved Diagnosis and Monitoring,, by Andreas Weinhaeusel et al., the authors aimed to elucidate antibody signatures in manifested and pre-diagnostic ulcerative colitis (UC) patients compared to controls using a high-content protein microarray.
Comments and Suggestions for Authors
Overall, this manuscript is well written and provides pertinent information that autoantibody profiling reveals early immune signatures in UC, providing novel biomarkers for preclinical diagnosis and disease monitoring. The study also showed that the overlap between pre-diagnostic and manifest UC antigenic profiles reinforces their biological relevance, linking them to molecular pathology. These findings highlight antibody profiling as an additional omics layer, paving the way for novel diagnostic and therapeutic strategies in UC management.
Reviewer has no major comments to this work.
Minor concerns
The article has some shortcomings:
Authors should double-check abbreviations and make the necessary corrections so that abbreviations are explained when they first appear, both in the abstract and in the manuscript text and figure legends.
*Introduction - Line 69, 70 - explain the abbreviations
*Subsection 2.2
- Table 2 is not found in the text of manuscript subsection 2.2.
- Figure 1 is not found in the text of manuscript subsection 2.2.
*Subsection 2.3
Table 3 is not found in the text of manuscript subsection 2.3.
Observation is also valid for the following Tables. Probably ,,Error! Reference source not found,, takes the place of Table.
References
The references do not follow the journal's recommendations:
- References should be described as follows, depending on the type of work:
Journal Articles:
- Author 1, A.B.; Author 2, C.D. Title of the article. Abbreviated Journal Name Year, Volume, page range.

Round 2
Reviewer 1 Report
Comments and Suggestions for Authors
Although manuscript ijms-3525991 is improved, some issues should be amended:
1. It has to be added in the Section 4.2 (at least) that approximation of IgG serum / plasma concentration was based on the concentration of purified IgG as all samples were processed in an identical way (same starting volume of the sample, same elution volume, etc.) and IgG subclass – specific interaction with matrix could be excluded.
Taking into account the above mentioned, I find acceptable approximation you made. However, that should be pointed out clearly. Melon™ Gel IgG Spin Purification Kit is declared to provide IgG recovery greater than 90% (not necessarily 100%, and not necessarily identical for all samples) with purity greater than 80% (particularly important when A280 is used for determination of protein concentration).
2. Legend to the Table 4 (lines 235 – 236, explanation for **) should be checked / corrected.
I suggest the following:
** increase in IgG concentration in T3 was evaluated with respect to the samples taken >12m before diagnosis, i.e. 9-24 months (median 13 months) before T3 (compared samples and their IgG concentrations are indicated by bold numbers in the time- and IgG concentration columns, respectively).
3. In line with previous suggestion related to Table 4, Figure 1, Prediagnostic part, should be corrected. Namely, selected T1/T2 plasmas (written in bold in Table 4) were collected 15 – 35 months (median 21 months) before UC diagnosis i.e. 9 – 24 months (median 13 months) before corresponding T3. Legend to the Figure 1 should be corrected accordingly as well (line 143). Further, in relation to volcano plots A and B, citation of https://doi.org/10.1371/journal.pone.0228615 has to be included in the legend to the Figure 1.

Author Response
Dear Reviewer,
Thank you for the thoughtful feedback, which helped us further improve our manuscript. We have edited our manuscript as outlined in the word file attached (see also the submitted manuscript with "changes tracked").
Thank you and kind regards, Andreas
